# Systematic Analysis of the Molecular Mechanisms of Cold and Hot Properties of Herbal Medicines

**DOI:** 10.3390/plants11070997

**Published:** 2022-04-06

**Authors:** Sang-Min Park, Su-Jin Baek, Hyo-Jeong Ban, Hee-Jeong Jin, Seongwon Cha

**Affiliations:** 1KM Data Division, Korea Institute of Oriental Medicine, Daejeon 34054, Korea; smpark@cnu.ac.kr (S.-M.P.); baeksj@kiom.re.kr (S.-J.B.); hjban@kiom.re.kr (H.-J.B.); hjjin@kiom.re.kr (H.-J.J.); 2College of Pharmacy, Chungnam National University, Daejeon 34134, Korea

**Keywords:** phytomedicine, cold pain, thermal hypersensitivity, norepinephrine, thermogenesis

## Abstract

Effective treatments for patients experiencing temperature-related symptoms are limited. The hot and cold effects of traditional herbal medicines have been utilized to treat and manage these symptoms, but their molecular mechanisms are not fully understood. Previous studies with arbitrarily selected herbs and ingredients may have produced biased results. Here, we aim to systematically elucidate the molecular mechanisms of the hot and cold properties of herbal medicines through an unbiased large-scale investigation of herbal ingredients, their target genes, and the transcriptome signatures induced by them. Using data regarding 243 herbs retrieved from two herbal medicine databases, we statistically identify (R)-Linalool, (-)-alpha-pinene, peruviol, (L)-alpha-terpineol, and cymol as five new hot-specific ingredients that share a common target, a norepinephrine transporter. However, no significant ingredients are cold-specific. We also statistically identify 14 hot- and 8 cold-specific new target genes. Pathway enrichment analysis of hot-specific target genes reveals the associated pathways including neurotransmitter reuptake, cold-induced thermogenesis, blood pressure regulation, adrenergic receptor signaling, and cation symporter activity. Cold-specific target genes are associated with the steroid pathway. Transcriptome analysis also shows that hot herbs are more strongly associated with coagulation and synaptic transmission than cold herbs. Our results, obtained from novel connections between herbal ingredients, target genes, and pathways, may contribute to the development of pharmacological treatment strategies for temperature-related pain using medicinal plants.

## 1. Introduction

Approximately one-fifth of the world’s population suffers from chronic pain [1]. Among the several kinds of chronic pain in existence, temperature-related pain (especially from coldness) is one of the most common symptoms [2], including thermal hyperalgesia (increased pain response to noxious temperatures) and thermal allodynia (pain response to innocuous temperatures) [3]. As this abnormal sensation is often most severe in the extremities, it is generally referred to as peripheral coldness [4], cold hypersensitivity in the hands and feet [5], or thermal discomfort with cold extremities [6,7]. Several putative mechanisms have been postulated to explain these symptoms, including dysregulation of temperature thresholds of voltage-gated ion channels and transient receptor potential ion channels [2,3,8], sensitization of C nociceptors or Aδ fiber neurons [3], impairment of thermogenesis and energy homeostasis [2], and abnormal cutaneous vasoconstriction [9]. Although the aforementioned symptoms have been found to reduce quality of life and cause excruciating pain [2,10], effective treatments for these conditions are limited.

Traditional herbal medicines utilize the properties of herbs with hot or cold temperament to treat and manage temperature-related pain [11]. Several studies have investigated the molecular mechanisms of the hot and cold properties of herbal medicines. Through network pharmacological and chemical fragment-based analyses, Liang et al. identified different pathways and target genes between 10 cold- 10 hot-temperament herbs [12]. While the hot property was associated with inflammation and immune regulation, the cold property was associated with cell growth, proliferation, and development. Zhou et al. identified the cold- and hot-specific metabolic profiles as well as the pathways of 7 cold- and 7 hot-temperament herbs using biochemical and metabolomic approaches [13]. The cold property was associated with promoting arginine and proline metabolism, tryptophan metabolism, and ATP storage and generation, while the hot property had the opposite effect. Based on the gene expression data previously obtained from MCF7 cells treated with herbal ingredients [14], Wang et al. identified the pathways and genes associated with the hot and cold properties based on the active ingredients of 6 cold- and 6 hot-temperament herbs [15]. Differences between cold and hot properties were enriched in the endocrine system, cell proliferation and apoptosis, and hemorheology. However, these studies have limitations in analyzing only small-scale, arbitrarily selected herbs and their associated ingredients.

To avoid biased results due to analysis of only a few specific herbs or ingredients, we aim to systematically elucidate the molecular mechanisms of the hot and cold properties of herbal medicines through an unbiased large-scale investigation of herbal ingredients, their target genes, and the transcriptome signatures induced by them. Using the data of 243 herbs obtained from herbal medicine databases, we perform statistical analysis to identify the enriched ingredients and target genes specific to herbs with hot or cold effects. The transcriptome signatures related to hot or cold effects are also analyzed from the gene expression data to reveal associated pathways. Our study provides a comprehensive understanding of the cold and hot properties of herbal medicine and possible therapeutic interventions for patients with temperature-related pain.

## 2. Materials and Methods

### 2.1. Extraction of Data from Herbal Medicine Databases

Information regarding herbal medicines was obtained from two recent databases: The Integrative database of Traditional Chinese Medicine enhanced by Symptom Mapping (SymMap; symmap.org accessed on 8 September 2020) [16] and the Encyclopedia of Traditional Chinese Medicine (ETCM; tcmip.cn/ETCM accessed on 7 September 2020) [17]. In these databases, the effects of herbs associated with a hot temperament are classified according to the following four levels: great/extreme hot, hot, warm, and mildly/minor warm. Similarly, the effects of herbs associated with a cold temperament are classified according to the following four levels: great/extreme cold, cold, cool, and mildly/minor/slightly cold. The effects of herbs that were neither hot nor cold were classified as even/mild.

Due to a parsing problem in ETCM, where commas in ingredient names are ambiguous because they are used for both synonym and chemical nomenclature punctuation, ingredient information for each herb was extracted from SymMap only. Data linked to the duplicate synonyms of each ingredient were integrated based on alias names, PubChem IDs, and CAS IDs, and then target gene information of the corresponding ingredient was extracted.

### 2.2. Enrichment Analysis for Ingredients and Target Genes

Hypergeometric tests were performed between the herb categories to identify the enriched ingredients or target genes associated with the hot or cold properties using the *dhyper* function in R. First, we collated ingredient information of all herbs in each category, and then calculated the frequency of occurrence of each ingredient in each category. Next, we also collated target gene information of all the ingredients of all herbs in each category, and then calculated the frequency of occurrence of each target gene in each category. The hypergeometric test was used to statistically compare the frequency of a particular ingredient or target gene in one category compared to another considering the overall size of the two categories. Adjusted *p* values, *P*_adj_, were obtained using Benjamini–Hochberg multiple testing correction after the hypergeometric tests were performed.

### 2.3. Pathway Enrichment Analysis

Using the identified specific target genes, we performed pathway enrichment analysis and network visualization [18]. Pathway enrichment analysis of the ranked gene list was performed using g:Profiler [19] against the gene sets of Gene Ontology (GO) terms for biological processes (BP), WikiPathways, and Reactome. The downloaded results were interpreted using the EnrichmentMap application [20] in Cytoscape. The enrichment analysis results were visualized as a network consisting of nodes representing statistically enriched pathways and links expressing similarities between nodes. Overlapping pathways formed clusters that were collapsible into single super-nodes, depicting major biological themes. Using this representation, we demonstrated the complex network of pathways as an organized map with default parameters (node cutoff of Q-value of 0.05 and an edge cutoff of similarity of 0.375).

### 2.4. Gene Expression Profile Analysis

To identify specific transcriptomic alterations associated with hot or cold properties, we obtained microarray data [14] containing the gene expression profiles of the MCF7 breast cancer cell lines treated with 102 ingredients of herbal medicines. The gene expression matrix was constructed using the *limma* package [21] after multi-array average normalization, and differential expression analysis was performed between the categorized ingredient groups. Gene set enrichment analysis (GSEA) was performed using the *fgsea* package [22] to analyze the gene expression profiles.

## 3. Results

### 3.1. Integration of Data from Databases of Herbs Annotated with Hot and Cold Properties

We integrated the information from databases to systematically investigate the molecular mechanisms underlying the cold and hot properties of herbal medicines. We obtained the annotated cold and hot properties of 322 herbs from SymMap [16] and 402 herbs from ETCM [17] (Figure 1a). We next integrated information from 258 common herbs between the two databases (Appendix A), and the effects of 214 herbs were exactly matched. Furthermore, 29 herbs were similarly matched, demonstrating the same hot or cold properties, albeit at different levels. In these cases, the herbs were classified based on the lower of the two values from each database. The remaining discordant 15 herbs were excluded from subsequent analysis.

The properties of the 243 matched herbs were categorized into one of nine temperament levels ranging from great/extreme cold to great/extreme hot (Figure 1b). They were then broadly categorized into one of three groups: the hot group (HG), cold group (CG), or even group (EG) for comparison. The herbs were further categorized as hot group extended (HGE), non-hot group (NHG), cold group extended (CGE), or non-cold group (NCG). In the subsequent analysis, we compared the ingredients of the herbs, their target genes, and the transcriptome signatures between the groups (Figure 1c).

### 3.2. Enrichment Analysis for Identification of Hot- and Cold-Specific Ingredients 

We first investigated the key ingredients contributing to the hot or cold properties of herbs. We performed enrichment analysis with the top 30 herb ingredients in the HG and CG using a hypergeometric test and Benjamini–Hochberg multiple testing correction (Appendix A). None of the ingredients in the CG were significantly enriched compared with those in the HG or NCG (*p*_adj_ > 0.05). The most significant differentially enriched CG ingredient was apigenin. However, 21 ingredients were significantly enriched in the HG compared with those in both CG and NHG (*p*_adj_ < 0.01) (Figure 2a). We identified the top five enriched ingredients in the HG as hot-specific ingredients: (R)-linalool, (-)-alpha-pinene, peruviol, cymol, and (L)-alpha-terpineol. The proportions of the hot-specific ingredients were higher in the HG than in the CG (Figure 2b). In particular, (R)-linalool and (-)-alpha-pinene were present in nearly 50% of the herbs in the HG. The ratio of peruviol in the HG was the lowest among the five hot-specific ingredients and was rarely observed in groups other than the HG and HGE.

To identify the molecular mechanisms of the hot-specific ingredients, we investigated their known target genes based on the databases (Appendix A). We identified solute carrier family 6 member 2 *(SLC6A2*), which encodes a norepinephrine (NE) transporter, as a common target gene of all hot-specific ingredients (Figure 2c), except (-)-alpha-pinene, whose target gene information was unavailable. In addition, *PTGS1* and *PTGS2* encoding prostaglandin-endoperoxide synthase 1 and 2, respectively, *CHRM1* encoding cholinergic receptor muscarinic 1, and *SCN5A* encoding sodium channel protein type 5 subunit alpha were also found to be targeted by the hot-specific ingredients.

### 3.3. Enrichment Analysis for Identification of Target Genes and Biological Pathways

Next, we investigated the enriched target genes of the herb ingredients in the HG and CG. We performed the enrichment analysis of the top 50 target genes using the hypergeometric test and Benjamini–Hochberg multiple testing correction (Appendix A). Each group contained significantly enriched target genes, and the overall significance level of the HG was lower than that of the CG (Figure 3a). The previously identified common targets of the hot-specific ingredients (*SLC6A2*, *CHRM1*, *PTGS2*, *PTGS1*, and *SCN5A*) were found to be commonly enriched across the HG. We defined the genes that were significantly more enriched in the HG than in either the CG or CGE (*p*_adj_ < 0.01) as hot-specific target genes, which were *CHRM1*, *SLC6A2*, *SLC6A3*, *SLC6A4, ADRA1A*, *PTGS2*, *RXRA*, *ADRB2*, *MAOA*, *MAOB*, *BCHE*, *ADRA2A*, *PLB1*, and *ADRB1*. Similarly, the cold-specific target genes, which were *NOS2*, *PRSS1*, *GSK3B*, *CALM3*, *CKD2*, *AR*, *CASP3*, and *ESR2,* were defined as the genes that were significantly more enriched in the CG than in either the HG or HGE (*p*_adj_ < 0.01). The full name of each target gene is provided in Appendix A. The respective proportions of the selected hot- and cold-specific target genes were higher and lower, respectively, in the CG than in the HG (Figure 3b). 

To demonstrate the hot and cold properties of the herbs at the level of biological pathways, we performed a pathway enrichment analysis of the hot- and cold-specific target genes using g:Profiler [19] with the GO BP gene set (Figure 3c). The cold-specific target genes were enriched in a few pathways associated with steroid hormones. In contrast, the hot-specific target genes were enriched in multiple pathways related to neurotransmitters, adrenergic signaling, and blood pressure. Through network analysis of the enrichment results of the hot-specific target genes using EnrichmentMap [18,20], we identified several functional clusters (Figure 4). Among these, the adrenergic receptor signaling pathway cluster was identified as the largest cluster, which served as a functional hub connecting pathway clusters regulating synaptic transmission, vasodilation, and blood pressure. Furthermore, the synaptic transmission cluster was also related to neurotransmitter metabolism and the activity of cation channels. Cold-induced thermogenesis was another identified cluster.

### 3.4. Transcriptomic Effects of Hot and Cold Ingredients

To examine the effects of the hot and cold properties at the transcriptome level, we analyzed the gene expression data obtained after treating MCF7 breast cancer cells with 102 herbal ingredients [14]. We calculated the proportions of the tested components in each herb category defined in this study (Figure 5a). Components with distributions skewed towards the CG or HG were selected for comparison. The following components, highly enriched in the HG, were named as hot components (HCs): isoborneol, honokiol, hypaconitine, curculigoside, hydroxysafflor yellow A, and cholic acid. The following components, highly enriched in the CG, were named as cold components (CCs): chlorogenic acid, ethyl caffeate, gentiopicroside, and artemisinin. We excluded anhydroicaritin from the CCs because it was previously classified as a hot component in a study [15].

To reveal the pathways differentially active between the HCs and CCs, we performed GSEA using the Hallmark and GO BP gene sets. Consistent with our previous findings, the upregulated genes after treatment with HCs were found to be associated with the downregulation of coagulation, which is related to vasodilation, blood pressure, and upregulation of synaptic transmission (Figure 5b). We also found that pathways involved in the upregulation of oxidative phosphorylation and cellular respiration were associated with HCs (Figure 5b).

## 4. Discussion

We systematically identified novel molecular mechanisms involved in the hot and cold properties of herbs from two herbal medicine databases (SymMap and ETCM) by investigating the key ingredients, target genes, and transcriptomic signatures. We found that 21 key ingredients were significantly enriched in the HG, with the top five ingredients defined as hot-specific ingredients. We also found that 14 hot-specific and 8 cold-specific target genes were significantly enriched in the HG and CG, respectively. In addition, consensus pathways related to synaptic transmission and blood pressure were identified from both pathway enrichment analysis of the target genes and transcriptome analysis of the gene expression data for herbal ingredients. Network analysis further suggested that adrenergic signaling and associated synaptic transport could be core pathways for the hot properties of herbal medicines.

(R)-Linalool, (-)-alpha-pinene, peruviol, (L)-alpha-terpineol, and cymol were identified as the top five hot-specific ingredients. (R)-Linalool is a natural monoterpene with anti-inflammatory and analgesic effects [23]. (-)-Alpha-pinene is also a monoterpene that has neuroprotective and anti-nociceptive effects [24]. Peruviol, also known as nerolidol and penetrol, is a naturally occurring sesquiterpene that reportedly has anti-nociceptive and anti-inflammatory properties [25]. (L)-Alpha-terpineol, another natural monoterpene, has an analgesic effect on neuropathic pain [26]. Cymol, also known as p-Cymene, is an alkylbenzene related to a monoterpene that exhibits analgesic, anti-nociceptive, vasorelaxant, and neuroprotective activities [27]. In summary, these hot-specific ingredients are commonly associated with pharmacological effects that reduce pain. The common target of these hot-specific ingredients was identified to be solute carrier family 6 member 2 (SLC6A2). Furthermore, single nucleotide polymorphisms in *SLC6A2* have been reported to be related to pain sensitivity [28], suggesting that SLC6A2 is a potential target of treatments for temperature-related pain.

Heat production is a function of adipose tissues induced by sympathetic neuron innervation from secreted catecholamines, such as NE. Exposure to low temperatures increases NE release from neurons, inducing lipid metabolism and thermogenesis in adipose tissue. Recent studies have shown that sympathetic neuron-associated macrophages regulate NE concentrations in adipose tissue by causing the reuptake of extracellular NE mediated by SLC6A2, the identified common target of hot-specific ingredients [29,30]. These imported NE molecules in sympathetic neuron-associated macrophages are then degraded by monoamine oxidase A (MAOA), another identified hot-specific target. These results suggest that herbs with hot properties may modulate the crosstalk between neurons and immune cells in adipose tissues by enhancing macrophage-mediated NE removal. Our pathway analysis also confirmed the involvement of the hot-specific target genes in cold-induced thermogenesis. The activity of adipose macrophages increases with aging and obesity, leading to a decline in catecholamine-induced lipolysis and a decreased response to cold and starvation [29,30]. Thus, herbs with hot properties may contribute to maintaining adipose macrophage activity against aging and obesity.

Heart rate and blood pressure are regulated by NE release. Previously, herbs with hot properties have been shown to stimulate NE release in PC12 cells, whereas herbs with cold properties had no effect or suppressed NE release [31]. This result is supported by our pathway enrichment analysis results, which showed that the hot-specific target genes were associated with arterial pressure and diameter, neurotransmission, and the adrenergic signaling pathway. We proposed that the hot-specific target genes, including *SLC6A2* and *MAOA*, may be involved in these effects of herbs with hot properties. In addition, the hot-specific target genes were also associated with cation symporter activity, consistent with the findings of a previous study, which showed that herbs with hot properties induced the Na^+^-K^+^-ATPase and Ca^2+^-Mg^2+^-ATPase pump activities [32]. Modulating the activity of ion channels may be another mechanism of action of hot herbs. A previous study reported that the transcriptomic alterations in neuronal models with cold allodynia were distinctly associated with the activities of ion channel neurotransmitter receptors compared with mechanical allodynia [8].

We found that the cold-specific ingredients, their target genes, and the associated pathways were less in number than their hot-specific counterparts. Similarly, a previous study examining 10 cold and 10 hot herbs identified 27 target protein molecules associated with hot properties and only 2 target protein molecules associated with cold properties [12]. These differences in numbers may be caused by hot herbs exhibiting several common mechanisms, such as those identified in our study, whereas cold herbs demonstrate heterogeneous mechanisms. Alternatively, this may be due to a higher number of studies focusing on hot herbs, resulting in more information on ingredients and target genes of hot herbs than cold herbs. Another possible explanation is that several treatments for cold symptoms have been developed throughout history, whereas the development of treatments for hot symptoms other than inflammation has been limited. Therefore, it is speculated that the mechanisms of herbs with hot properties converge on common pathways, since the historical practices and development of treatment strategies have focused on cold symptoms. Moreover, the target genes in the steroid pathway may appear to be common in the herbs with cold properties, because inflammation is a major and prevalent cause of hot symptoms.

The molecular response of a cell to treatment with specific ingredients or drugs can be systematically assessed by its transcriptomic signature, which refers to the changes induced in gene expression after treatment. Therefore, we analyzed the gene expression data for the molecular responses against 102 herbal ingredients [14]. This dataset was generated in the breast cancer cell line MCF-7, which is not an optimal model for representing our body’s molecular response to hot or cold herbs. However, this MCF-7 dataset was the only available dataset that provided large-scale transcriptome responses to herbal ingredients. We should note that further research is needed to experimentally confirm the proposed molecular mechanisms of the cold and hot properties of herbal medicines. Despite such limitations, we identified consensus pathways related to synaptic transmission and blood pressure specifically affected by the hot-herb-enriched ingredients compared with the cold-herb-enriched ingredients. 

A previous study also analyzed the same dataset under the assumption that the major ingredients of herbs are representative of their effects [15]. However, a herb contains a variety of components in addition to its major ingredients. The interactions between these ingredients may lead to complex molecular responses through synergism, additivity, or antagonism, owing to nonlinear dynamics arising from feedback and crosstalk in biomolecular networks [33,34]. As a result, the actual activities of a herb resulting from all ingredients collectively may differ from the activities of the individual ingredients. The cold or hot properties of herbs can also be determined by examining the effect of combinations of ingredients. The effect of the different ingredient content of each herb should also be considered. To comprehensively analyze all these effects, transcriptome data generated through experiments using numerous herbal ingredients, their combinations, and herb extracts themselves are required. However, a well-organized herb-level transcriptome database has not yet been established. Although most existing databases for herbal medicine, including SymMap and ETCM, provide information on target genes of ingredients and herbs, they lack information on the respective molecular responses in cells. Only a recent database, HERB, provides a limited set of herb-level transcriptome data obtained from experiments under heterogeneous contexts and conditions [35]. Baek et al. recently demonstrated that systematic transcriptome analysis for *Paeoniae Radix* can reveal novel mechanisms of action [36]. Therefore, the development of a large-scale herb-level transcriptome database constructed through standardized experiments will significantly contribute to systematic studies of the molecular mechanisms of herbal medicines.

## 5. Conclusions

We conclude that the novel molecular mechanisms underlying the effects of herbs with hot properties are associated with multiple pathways including NE synaptic transmission, adipose tissue thermogenesis, and blood pressure regulation. Their pharmacological activities may act on major protein targets such as SLC6A2 and MAOA. Although the molecular mechanisms underlying the effects of herbs with cold properties appear heterogeneous, the steroid hormone signaling pathway was found to be a common mechanism among them. Our findings, based on the comprehensive systematic approach, may contribute to the development of treatment strategies for temperature-related pain.

## Figures and Tables

**Figure 1 plants-11-00997-f001:**
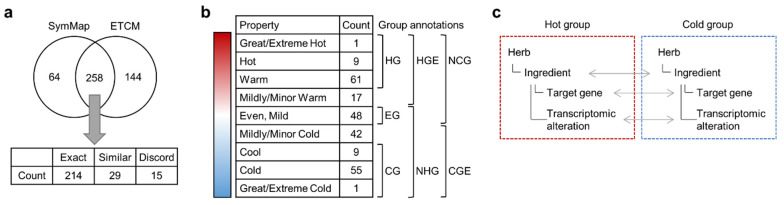
Integration of databases for the analysis of hot and cold herbs. (**a**) Comparison of herbs and their cold/hot properties obtained from the following two databases: SymMap and ETCM. (**b**) Summary table of integrated properties of 243 herbs that were similar or exact matches. Herbs were classified as a hot group (HG), cold group (CG), or even group (EG) based on their properties. Herbs were further classified as hot group extended (HGE), non-hot group (NHG), cold group extended (CGE), or none-cold group (NCG). The counts represent the number of herbs classified into each group. (**c**) Comparison of hot and cold groups at different levels.

**Figure 2 plants-11-00997-f002:**
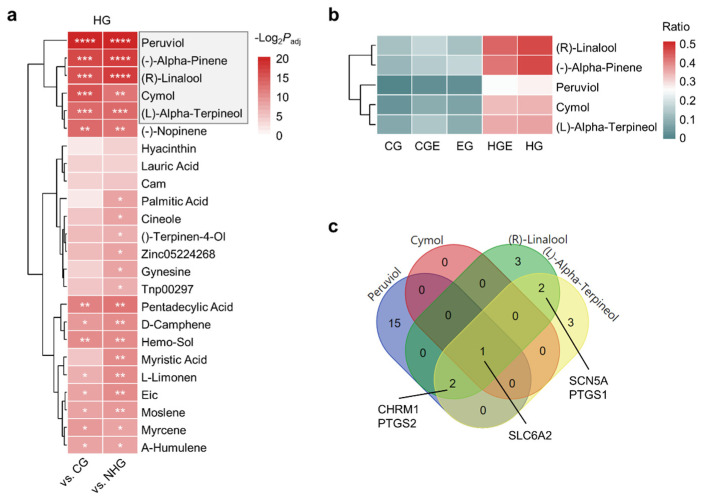
Enriched ingredients of herbs in the hot group. (**a**) Enrichment analysis of ingredients comparing the herbs of the HG with those of the CG and NHG. The *P*_adj_ values were obtained by Benjamini–Hochberg multiple testing correction after hypergeometric tests. The top five hot-specific ingredients are denoted with the gray box. * *p*_adj_ < 10^−2^, ** *p*_adj_ < 10^−3^, *** *p*_adj_ < 10^−4^, **** *p*_adj_ < 10^−5^. (**b**) Ratio of hot-specific ingredients included in the herbs of the CG, CGE, EG, HE, and HG. (**c**) Venn diagram of the target genes of the hot-specific ingredients. The common target genes are indicated in their groups using lines. Lists of all the target genes are provided in Appendix A.

**Figure 3 plants-11-00997-f003:**
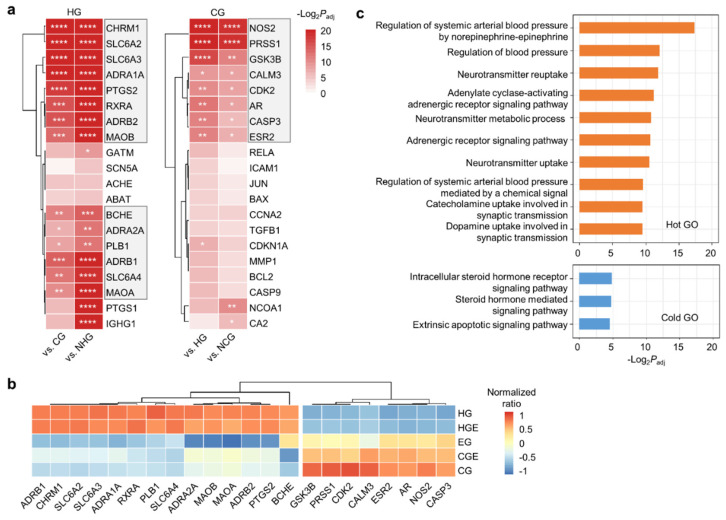
Enriched target genes of herbs in the hot and cold groups. (**a**) Enrichment analysis comparing the target genes of the HG with those of the CG and NHG (left) as well as comparing the target genes of the CG with those of the HG and NCG (right). The *p*_adj_ values were obtained by Benjamini–Hochberg multiple testing correction after hypergeometric tests. The hot- and cold-specific target genes are denoted with the gray boxes. * *p*_adj_ < 10^−2^, ** *p*_adj_ < 10^−3^, *** *p*_adj_ < 10^−4^, **** *p*_adj_ < 10^−5^. (**b**) Normalized ratios of the top five hot- or cold-specific target genes of the herbs of the CG, CGE, EG, HE, and HG, calculated as z-scores. (**c**) Pathway enrichment analysis of the target genes of the HG (upper panel) and CG (lower panel) using g:Profiler with the GO BP gene set. The top 10 functional characteristics were plotted, with *p*_adj_ < 0.05.

**Figure 4 plants-11-00997-f004:**
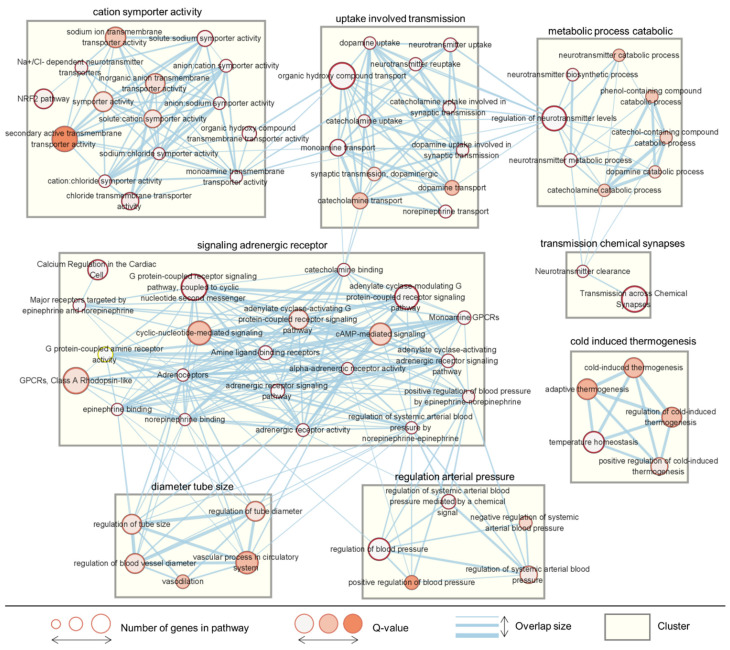
Network analysis of the enriched pathways in the target genes of the HG. A representative functional map of the associated pathways was constructed using EnrichmentMap. Nodes represent the enriched pathways and links represent the similarities between them. The size and color of the nodes indicate the number of genes included in the pathway and the statistical significance of the Q-value, respectively. The thickness of the link indicates the degree of overlap of the genes between the two connected pathways. The rectangular box indicates the identified cluster.

**Figure 5 plants-11-00997-f005:**
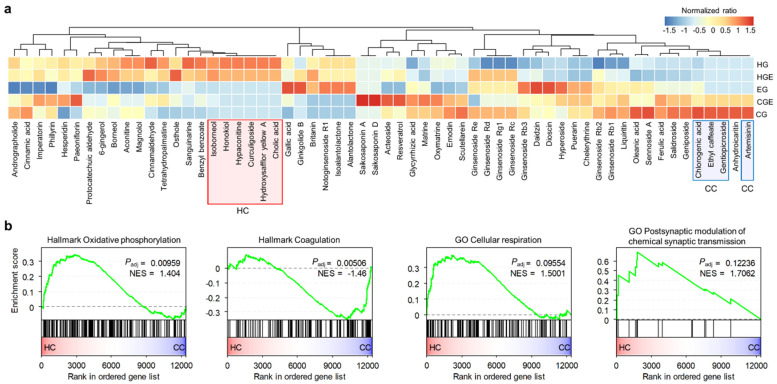
Transcriptomic analysis of the ingredients of the hot and cold groups. (**a**) Normalized ratios of ingredients included in the CG, CGE, EG, HE, and HG of the analyzed gene expression data, calculated as z-scores. The selected hot components (HC) and cold components (CC) are denoted with red and blue boxes, respectively. (**b**) Comparison of the GSEA results of HC and CC using Hallmark and GO BP gene sets. NES; normalized enrichment score.

## Data Availability

All data generated or analyzed during this study are included in this published article (and its Appendix A).

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
