# Peer review of "Systematic Analysis of the Molecular Mechanisms of Cold and Hot Properties of Herbal Medicines"

_plants, 2022, doi:10.3390/plants11070997_

Round 1
Reviewer 1 Report
The manuscript entitles “Systematic analysis of the molecular mechanisms of cold and hot properties of herbal medicines” investigated the molecular mechanism of cold and hot resistance. The subject frame of the work is well constructed. So, in this respect and this article should be contributed to present research. I recommended this work for publication after the following minor revisions.
- There are several typographical mistakes as well in whole manuscript. Therefore, the author’s thoroughly careful check the language and typo mistake to minimize the error.
- The abstract should be beginning with a sentence about the background of concept and the aims as well as novelty of study should be mentions. What exactly is the novelty of this study? The abstract is poorly written and should be improved. Abbreviations must be avoided in abstract. Parenthesis should be avoided in abstract - this is poor writing. Please improve.
- Introduction; Check and format the citations in the whole manuscript. Also, Appropriate references must be provided to explained the background, what is already done and why this study carried out. Other vise the novelty of this research is still poorly presented. This is important especially for the high IF journals. The scientific style should be used. What exactly is the aim of this work? Hypothesis statement is missing in the introduction section.
- Results and discussion; General remark to the discussion - In my opinion, the discussion provided by Authors is difficult to follow and verify due missing critical details in the methodology section. Due to poorly described material and poorly presented methods, I am not able to follow and properly review the discussion.
- All figures are of poor technical quality and not suitable for publication, especially in a high reputed journal. Font size and kind is too small and must be unified in all figures. Small writings are unreadable. All figures must be self-explanatory. Axis titles are poorly presented or absent. Units are missing. Are the data presented in figures significantly different? At least error bars should be shown.
- I suggest first time write full name rather than abbreviation; revise throughout in manuscript
Reviewer 2 Report
The authors compare 243 herbs integrated from two herbal medicine databases due to hot and cold properties, as well as their potential molecular mechanisms. The presented research is very interesting and we can see the current possibilities of collecting data in various databases are huge. The authors used these databases to elucidate potential mechanisms at the compound composition level and the relevant molecular mechanisms. I appreciate the enormous amount of work put into collecting all the data in the publication, however, I have a few questions regarding the presented results.
Please explain how was analysed ingredients from particular plants?
For example: Croton Fruit on ETCM (id 4) are shown more than 50 components and also Symmap (id 6) show more than 50 ingredients from Croton fruit. Did you used all compounds from those two database and compare all of them or only few (most common) ?
How those ingredients name was transfer for statistic analysis, as well as targed genes?
More details are needed .
The authors compare 243 herbs integrated from two herbal medicine databases due to hot and cold properties, as well as their potencial molecular mechanisms. The presented research is very interesting and we can see the current possibilities of collecting data in various databases are huge. The authors used these databases to elucidate potential mechanisms at the compound composition level and the relevant molecular mechanisms. I appreciate the enormous amount of work put into collecting all the data in the publication, however, I have a few questions regarding the presented results.
Please explain how was analysed ingredients from particular plants?
For example: Croton Fruit on ETCM (id 4) are shown more than 50 components and also Symmap (id 6) show more than 50 ingredients from Croton fruit. Did you used all compounds from those two database and compare all of them or only few (most common) ?
How those ingredients name was transfer for statistic analysis, as well as targed genes?
More details are needed .
Reviewer 3 Report
The paper seems interesting but its reading rises many questions about the methodology used.
The Materials and Methods description is minimal and would certainly not allow the reproduction of such experiments.
How did you determine the main phytochemicals associated for each herbal preparation (database, your analyses?). Most of these phytochemicals are terpenes which are know to widely varied according to the culture and/or meteorological conditions, genotypes, etc. Did you take into account the relative content of each phytochemical for each herbal preparation? How did you take into account the possible synergistic effect between different phytochemicals?
The legend of Figure 2C is unclear and incomplete. Please revise it.
I am not convinced that MCF-7 cancer cell line is the best model to identified hot or cold associated gene expression in an unbiased way.
Did you performed a validation, for example using one of the identified compounds to validate the proposed action?
Defined the abbreviations for each target genes.
Avoid splitting legends from the figure itself on 2 separate pages.
Round 2
Reviewer 3 Report
The Authors have replied to all my questions and consideres all my suggestions. I recommend the present paper for publication in the present revised form.